# Cost-Effective Hyperparameter Optimization for Large Language Model Generation Inference

**Chi Wang**[1]  **Susan Xueqing Liu**[2]  **Ahmed H. Awadallah**[1]

[1]Microsoft Research, Redmond
[2]Stevens Institute of Techology

**Abstract**  Large Language Models (LLMs) have sparked significant interest in their generative capabilities, leading to the development of various commercial applications. The high cost of using the models drives application builders to maximize the value of generation under a limited inference budget. This paper presents a study of optimizing inference hyperparameters such as the number of responses, temperature and max tokens, which significantly affects the utility/cost of text generation. We design a framework named EcoOptiGen which leverages economical hyperparameter optimization and cost-based pruning. Experiments with the GPT-3.5/GPT-4 models on a variety of tasks verify its effectiveness. EcoOptiGen is implemented in the "autogen" package of the FLAML library: `https://aka.ms/autogen`.

## 1 Introduction

Large language models (LLMs) like GPT-3.5 and GPT-4 (Brown et al., 2020; Ouyang et al., 2022; OpenAI, 2023) have demonstrated impressive capabilities in a wide range of generative tasks, including story telling (Lucy and Bamman, 2021; Chen, 2022), code generation (Trummer, 2022; Poesia et al., 2022), math problem solving (Cobbe et al., 2021; Zong and Krishnamachari, 2022), and many others (Wang et al., 2022). Even though the LLMs do not always generate perfect answers, they have initiated a trend of building powerful user experiences such as coding assistants and chat-enabled search engines. As the interest of building LLM-enabled applications keeps growing, the demand for technologies for getting the best value out of generation inference will also grow.

The research community has recently studied the effect of individual hyperparameters on the inference performance, such as the prompt (Liu et al., 2021; Mishra et al., 2021; Shieh, 2022) and temperature (Branwen, 2020; Nadeem et al., 2020). However, little is known about how to optimize the different hyperparameters collectively and systematically. Moreover, the monetary cost is a concern for most application builders. High costs and implications on energy consumption and environmental impact (Schwartz et al., 2020) provide a strong incentive to systematically optimize the hyperparameters towards maximal utility and minimal cost.

In this paper, we present the first study on the systematic hyperparameter optimization for text generation inference using LLMs. Given the cost concern, we adopt an economical hyperparameter optimization method (Wang et al., 2021), and propose a cost-based pruning strategy to improve the optimization efficiency under budget constraints. We apply our optimization framework, named *EcoOptiGen*, to tune multiple hyperparameters jointly, including the number of responses, max tokens, temperature, probability mass, and prompts.

To study the effectiveness of EcoOptiGen, we evaluate it on the following datasets: APPS (Hendrycks et al., 2021a), HumanEval (Chen et al., 2021) (for code generation); MATH (Hendrycks et al., 2021b) (for math problem solving); and XSum (Narayan et al., 2018) (for text summarization). On all the four datasets, we observe that EcoOptiGen can find higher quality hyperparameter settings than the default settings suggested by a recent LLM benchmark HELM (Liang et al., 2022) or simple modifications for the same budget. Our pruning technique is

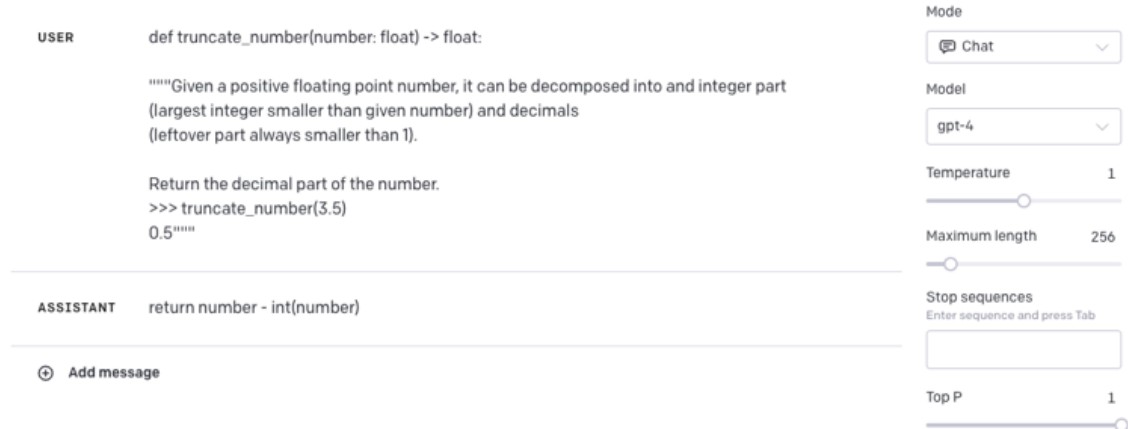

Figure 1: A code generation example using GPT-4, and a few examples of hyperparameters users can set for the inference, such as prompt, temperature, and max tokens.

shown to be effective in increasing the tuning performance significantly. We further find that the holistic hyperparameter optimization can mitigate idiosyncrasies to prevent suboptimal results.

## 2 Background

### 2.1 Text Generation with LLMs

**The Input and Output of LLM Text Generation**. Figure 1 shows one example of the input prompt to LLM. Upon receiving the prompt, LLM performs inference to generate one or more output responses. The input prompt can further include multiple examples to demonstrate what kind of responses are desirable. The output can be consumed or validated by an application in various ways, e.g., code executor (Hendrycks et al., 2021a) or math expression checker (Hendrycks et al., 2021b). It is often helpful for the inference to generate multiple responses and search for the best one, e.g., in code generation (Chen et al., 2021) and machine translation (Wullach and Chazan, 2022). The utility of the generated text is determined by the application consuming it. For example, when a predefined programmatic test is provided before generation, the code generation can be considered as successful if one of the generated responses can pass the test.

**The Cost of Text Generation with LLMs**. The cost of using LLMs for a text generation request is proportional to the amount of computations required to generate the output. The amount of computations is determined by the number of tokens in both the input and output. LLMs are often used as a service and charged based on the usage. From the perspective of an application builder using LLMs, the goal is to maximize the utility of the generated text under an inference budget constraint (e.g., measured by the average dollar cost needed to solve a coding problem). This can be achieved by optimizing the hyperparameters of the inference, which can significantly affect both the utility and the cost of the generated text. In the next section, we will discuss how hyperparameters affect the utility and the cost.

### 2.2 How Do Hyperparameters Affect Text Generation Performance?

**The Impact of Individual Hyperparameters**. We take a representative API from OpenAI, i.e., the completions API (ope, 2023), to analyze the tunable hyperparameters and their impact on the cost and the utility (e.g., accuracy, success rate): (1) model - this is a required input, specifying the model ID to use. (2) prompt - the input prompt to the model, which provides the context for the text generation task. (3) max_tokens - the maximum number of tokens (words or word pieces) to

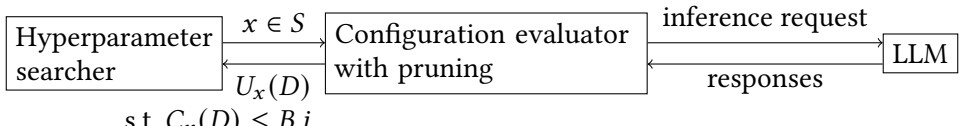

generate in the output. (4) temperature - a value between 0 and 1 that controls the randomness of the generated text. A higher temperature will result in more random and diverse text, while a lower temperature will result in more predictable text. (5) top_p - a value between 0 and 1 that controls the sampling probability mass for each token generation. A lower top_p value will make it more likely to generate text based on the most likely tokens, while a higher value will allow the model to explore a wider range of possible tokens. (6) n - the number of responses to generate for a given prompt. Generating multiple responses can provide more diverse and potentially more useful output, but it also increases the cost of the request. (7) stop - a list of strings that, when encountered in the generated text, will cause the generation to stop. This can be used to control the length or the validity of the output. (8) presence_penalty and frequency_penalty - values that control the relative importance of the presence and frequency of certain words or phrases in the generated text. These hyperparameters can be useful for controlling the focus and balance of the generated text. (9) best_of - the number of responses to generate server-side when selecting the "best" (the one with the highest log probability per token) response for a given prompt.

**The Joint Impact of Multiple Hyperparameters**. It can be seen that the cost and utility of text generation are intertwined with the joint effect of these hyperparameters. There are also complex interactions among subsets of the hyperparameters. For example, the temperature and top_p are not recommended to be altered from their default values together because they both control the randomness of the generated text, and changing both at the same time can result in conflicting effects (ope, 2023); n and best_of are rarely tuned together because if the application can process multiple outputs, filtering on the server side causes unnecessary information loss; both n and max_tokens will affect the total number of tokens generated, which in turn will affect the cost of the request. These interactions and trade-offs make it difficult to manually determine the optimal hyperparameter settings for a given text generation task.

## 3 EcoOptiGen

We first introduce the following notations and definitions for EcoOptiGen:

**Tuning Data** $D$, a small set of examples which can be used to measure the goodness of each hyperparameter configuration. Each data example contains a few text fields. For example, the HumanEval (Chen et al., 2021) dataset contains an input field which is the concatenation of the Python function signature and the doc string, and a test field of the test code.

**Utility Function** $U$, a function that represents the utility (a real-valued score) of the generated text. For example, for code generation, the utility is the success rate of passing the test; for text summarization, the utility is the effectiveness of summarization such as the rouge score. The utility of multiple verifiable responses (i.e., the best response can be selected from them by the application) is defined as the best utility score in all responses.

**Budget Constraints** $B = (B.i, B.o)$, a tuple of two values: $B.i$ is the average inference budget per example in the tuning data $D$ and $B.o$ is the total optimization budget. Generally, they are measured as the dollar cost which is proportional to the number of tokens in the input and output (ref. Section 2). When the price per token is a constant for both input and output, the dollar cost can be converted to the number of tokens in the input and output. For simplicity of illustration,

we use the number of tokens as the notion of budget in this section: $B.i$ is the allowed average number of token consumptions per request, such as 1K, and $B.o$ is the the total number of tokens allowed to consume in the tuning process, such as 1M. When $B = (1K, 1M)$, $|D| = 20$, if each request consumes exactly 1K tokens, the number of allowed hyperparameter configurations to try during the optimization is equal to $1M/1K/20 = 50$.

**Search Space** $S$, a dictionary where the key is each hyperparameter's name and the value is the range of values to search for the corresponding hyperparameter. Following the analysis in Section 2.2, we design a default search space as shown in Table 1, if users do not provide this input. The necessity of tuning certain hyperparameters depends on the application.

Table 1: Default search space for the optimization framework. Some hyperparameters are fixed as constants by default but users can override all of them according to domain knowledge.

| Hyperparameter | Default search range | Note |
| --- | --- | --- |
| model | ["text-ada-001", "text-babbage-001", "text-davinci-003", "gpt-3.5-turbo", "gpt-4"] | GPT models with diverse cost-quality trade-off |
| prompt | ["{*prompt*}"] | A list of prompt templates. "{*prompt*}" will be replaced by a input field named "prompt" in each instance to produce the actual prompt per instance. Typically overridden by users based on domain knowledge. Users can also use this list to choose among different prompting strategies, such as whether to use chain-of-thought or in-context-learning examples. |
| max_tokens | lograndint(100, 1000) | Random integers, logarithm distributed |
| temperature_or_top_p | [{"temperature": uniform(0, 1)}, {"top_p": uniform(0, 1)}] | Hierarchical search space: one configuration will either choose a temperature or a top_p |
| n | randint(1, 100) | Random integers between 1 and 100 |
| stop | None | Users can specify application-dependent stop choices |
| presence_penalty | 0 | Users can specify a float range within [-2, 2] if needed |
| frequency_penalty | 0 | Users can specify a float range within [-2, 2] if needed |
| best_of | 1 | Users can specify an int range lower bounded by 1 and fix n to 1 |

**Average Utility and Cost Consumptions for Configuration** $x$ **over Instances in** $D$: $U_x(D)$ and $C_x(D)$. A configuration $x$ is *invalid* if $C_x(D) > B.i$.

Within the specified optimization budget $B.o$, the framework iteratively tries different configurations in the given search space $S$, and outputs a configuration $x^*$ with the maximal average utility $U_{x^*}(D)$ on the tuning data $D$ subject to the average inference budget $C_{x^*}(D) \leq B.i$. The architecture is depicted in Figure 2. A *hyperparameter searcher* proposes configurations, and it invokes a *configuration evaluator* to assess the validity and utility of each configuration.

### 3.1 Hyperparameter Searcher

We opt for a blackbox optimization approach because we aim to make the framework generically applicable to (1) LLMs as a service, (2) complex utility functions which potentially involve blackbox evaluation of the output returned by LLMs. Our framework abstracts away from the internal process of computing the utility for a configuration, which involves making requests to LLMs and evaluating the responses with application-specific procedures.

There are a variety of blackbox optimization techniques, such as random search (Bergstra and Bengio, 2012), Bayesian optimization (Bergstra et al., 2011), evolutionary search (Goldberg and Deb, 1991), and local search (Wu et al., 2021). We chose a method that combines Bayesian optimization and local search, named BlendSearch, due to its cost efficiency (Wang et al., 2021). The local search method in BlendSearch performs randomized direct search with a provable convergence rate and cost bound. Bayesian optimization is used to generate starting points for the local search, and

different local search threads are prioritized adaptively. The original BlendSearch is designed for training hyperparameter optimization and used the training time as the measurement for cost. We adapt it to optimize the inference hyperparameters, and generalize the cost metric.

## 3.2 Configuration Evaluator

A simple evaluator takes a configuration $x$ as the input and outputs the metric to optimize. It loops over the data examples in $D$. For each example $d_i \in D$, a request is made to the LLM service using the configuration and the input fields $d_i.in$. The responses from the LLM service are then used to compute the utility $U$, and measure the cost consumption $C$. When the loop is over, the average cost $C_x(D)$ is compared with the user-provided bound $B.i$. If $C_x(D) \leq B.i$, the average utility $U_x(D)$ for all the data in $D$ is returned, otherwise a zero value is returned to indicate that the configuration is invalid. In the following, we present an improvement to the simple evaluator.

If a configuration $x$ is invalid, it is beneficial to terminate the trial early to save unnecessary cost. We design a pruning strategy by judiciously varying two cost-related factors during a trial: the number of tuning data examples, and the number of responses. A full evaluation of a configuration $x$ needs to send $|D|$ LLM requests, each asking for $x.n$ responses, where $x.n$ is the setting of the hyperparameter n in configuration $x$ (or the setting of best_of if best_of is searched instead of n). Our goal is to spend a much smaller cost in invalid trials. The full procedure is detailed in Algorithm 1 in the appendix.

**Initial Validity Check**. For a given configuration $x$, before the expensive evaluation starts, we first check whether we could prune the configuration directly (line 4-10 of Algorithm 1). This check is based on an assumption specific to our optimization problem.

**Assumption 3.1.** Given two configurations $x_1$ and $x_2$ with the same setting of model, prompt, and stop, if the number of responses and max_tokens in $x_1$ are both equal or larger than those in $x_2$, then we expect $x_1$ has an equal or higher average token consumption than $x_2$.

A consequence of the assumption is that if $x_2$ is invalid, then $x_1$ is invalid too. If $x_1$ is valid, then $x_2$ is valid too. Our pruning leverages this assumption to find a max known valid $n$ and a min known invalid $n$ for a configuration $x$, using the valid and invalid sets of already tried configurations $X_{\text{valid}}$ and $X_{\text{invalid}}$ which share the same setting of model, prompt and stop with $x$.

$$\text{max\_valid\_n} = \max_{x' \in X_{\text{valid}}, x'.\text{max\_tokens} \geq x.\text{max\_tokens}} x'.n \tag{1}$$

$$\text{min\_invalid\_n} = \min_{x' \in X_{\text{invalid}}, x'.\text{max\_tokens} \leq x.\text{max\_tokens}} x'.n \tag{2}$$

Then, depending on the relation among max_valid_n, min_invalid_n, and $x.n$, we do the following:

1. If $x.n \leq$ max_valid_n, we evaluate this trial as is. This corresponds to the case $x$ is expected to be valid based on Assumption 3.1.

2. Otherwise ($x.n >$ max_valid_n), if $x.n \geq$ min_invalid_n, we prune this trial without any evaluation. This corresponds to the case where $x$ is expected to be invalid based on Assumption 3.1.

3. Otherwise (max_valid_n $< x.n <$ min_invalid_n), we start evaluating the trial from number of responses equal to max_valid_n. This corresponds to the case where $x$ is expected to be either valid or invalid, and max_valid_n is expected to be a valid number of responses to use for $x$.

The order of our check takes into the consideration that Assumption 3.1 can be violated occasionally, i.e., max_valid_n may be occasionally equal or larger than min_invalid_n. By checking condition 1 before condition 2, we keep the chance of evaluating a trial when the violation happens.

**The Outer Loop of Algorithm 1: Varying the Number of Responses.** After the initial check is passed (case 1 or 3), we evaluate the configuration by gradually doubling the number of responses until it reaches $x.n$ (line 11-end). The starting point of $n$ is decided according to the rules above. To evaluate a particular $n$, we temporarily modify $x.n$ as $n - n'$, where $n'$ is the last evaluated number of responses for the current configuration in $D$ (0 at the start point or when data skipping happens, as explained in the next paragraph). That makes use of the results from requests made for evaluating smaller $n$ for the current configuration and saves cost compared to requesting $n$ responses. Note that while the gradual increase of $n$ makes it possible to terminate a trial with a smaller number of responses, it can also increase the total cost of evaluating a valid trial as the input tokens occupy the consumption in every request repeatedly. That issue is mitigated by a few choices in our design: (a) we start from the max known valid $n$ instead of 1; (b) the geometric increase of $n$ reduces the number of times that $n$ is varied, to a logarithm factor, rather than a linear factor in a linear schedule, and (c) the data skipping described next helps reducing the number of requests for smaller $n$ if the trial is valid.

**The Inner Loop of Algorithm 1: Varying the Number of Data Examples.** For each fixed number $n$ of responses, we employ progressive subsampling (Provost et al., 1999) over the tuning data $D$ to prune a trial (line 12-32). After we get the responses for $k$ examples in $D$, we can compute the mean of their cost and utility. We denote the subset of the $k$ examples as $D_k$. $C_x(D_k)$ is an estimate of $C_x(D)$. Hoeffding-Serfling inequality (Bardenet and Maillard, 2015) can be used to compute the upper (lower, resp.) bound for $C_x(D_k)$ if $C_x(D)$ is indeed below (above, resp.) $B.i$. If $C_x(D_k)$ is larger than the upper bound, we terminate the trial, and update $X_{\text{invalid}}$. If $C_x(D_k)$ is smaller than the lower bound and the current $n$ is smaller than the original $x.n$, we skip the remaining data points in $D$, reset $n' = 0$, and update $X_{\text{valid}}$. The number $k$ is doubled until it reaches $|D|$. The geometric increase of $k$ limits the number of times this hypothesis test is conducted per trial to reduce the chance of incorrect pruning.

## 4 Experiments

We are interested in investigating the following research questions: First, for text generation tasks, how much gain can EcoOptiGen achieve by tuning the hyperparameter settings under a budget constraint? Second, how does varying the inference budget affect the optimization result? Third, how does varying the model affect the optimization result? In this section, first, we describe the setting of our experiment in Section 4.1. Then, we investigate the three research questions in Section 4.2 through 4.4. We discuss limitations and future work in Section 4.6.

### 4.1 Setup

**Datasets.** To evaluate the performance of EcoOptiGen, we select a diverse set of text generation tasks from the HELM benchmark (Liang et al., 2022) (v1.0): code generation, math problem solving, and text summarization. For code generation, we evaluate EcoOptiGen on two datasets: APPS (Hendrycks et al., 2021a) (a dataset for generating Python code for a coding problem given the problem description) and HumanEval (Chen et al., 2021) (a dataset for generating Python code based on the function name and docstring). For math problem solving, we use the MATH dataset (Hendrycks et al., 2021b) (a dataset for math problems containing chain-of-thoughts, i.e., the derivation steps for the solution). For text summarization, we evaluate EcoOptiGen's performance on the XSum dataset (Narayan et al., 2018) (a large dataset for summarizing news articles).

For tuning, we randomly sample 20 examples for tuning of all the datasets, except for 60 of XSum since it contains more data. For each dataset, we randomly sample a few hundred examples for testing. The test set selection procedure follows HELM. For MATH, we use "Level 1" problems in Section 4.2 to 4.4 following HELM.

Table 2: Results using 'best' GPT-3.5 model (among code-davinci-002, text-davinci-002, and text-davinci-003) according to HELM.

| Method | APPS | HumanEval | MATH | XSum |
|---|---|---|---|---|
| HELM | 0.03 | 0.465 | 0.378 | 0.140 |
| EcoOptiGen (HELM budget) | 0.05 | 0.521 | 0.414 | 0.144 |
| Search | 0 | 0.493 | 0.769 | 0.136 |
| Search+PSR | 0 | 0.493 | 0.739 | - |
| EcoOptiGen | 0.05 | 0.792 | 0.771 | 0.144 |
| HELM (modified) | 0.03 | 0.701 | 0.403 | 0.140 |

**Evaluation Metric.** For the two code generation tasks, test cases are already provided by the datasets for verifying responses at inference time. For each code generation instance, as long as any response passes the test cases, the score that EcoOptiGen receives for this instance is 1 and 0 otherwise. For MATH, we consider two ways of evaluation. In Section 4.2 to Section 4.4, we define "success" as: if one of the returned chain-of-thought responses has an equivalent final answer with the ground truth, EcoOptiGen receives 1 for this instance and 0 otherwise. In Section 4.5, we define "success_vote" as: if the response based on majority voting has an equivalent final answer with the ground truth, EcoOptiGen receives 1 for this instance and 0 otherwise. For XSum, we use 'best_of' to rerank the generated responses by their mean log probabilities and use the Rouge-2 score for the top response (Lin, 2004).

**Comparative Methods.** We evaluate the following alternatives to compare with EcoOptiGen:
 • HELM. We check the best score evaluated under the HELM benchmark. For each dataset, we find the best performing GPT-3.5 model (code-davinci-002, text-davinci-002, and text-davinci-003) reported by HELM, and then re-evaluate on our test data using the same model and hyperparameter settings (with modified prompts on APPS and MATH as explained in the next paragraph) from HELM. The complete details of these configurations are listed in Table 4 of the appendix. The reason to use this baseline is that HELM has a broad coverage of the latest LLMs with a specific hyperparameter setting per task, which is rare to find elsewhere.
 • Search. This is a method that applies the same hyperparameter searcher as EcoOptiGen but does not use pruning or alter the optimization metric.
 • Search + PSR. This is the method that is the same as Search, but uses probabilistic success rate instead of success rate as the optimization metric. Not relevant in the XSum dataset.
 By default, the input to all the search-based methods is set to $B.i = 1K, B.o = 1M$. The search space follows Table 1, while "model" and "stop" are overriden by the HELM config. For HumanEval, we search the prompts over four templates: "{definition}", "# Python 3{definition}", "Complete the following Python function:{definition}", and "Complete the following Python function while including necessary import statements inside the function:{definition}". For the purpose of saving inference cost, we use zero-shot prompt rather than the two-shot used in HELM on APPS. For MATH, we use only one fixed demonstration example for all categories in the prompt as opposed to eight per category in HELM. For XSum, the same prompts, n and max_tokens from HELM are used, while best_of is searched in the range of randint(1, 100); and the budget is set to $B.i = 2K, B.o = 4M$.

## 4.2 EcoOptiGen's Performance

The performance scores of EcoOptiGen are shown in Table 2 along with other comparative methods. For all the 4 datasets, EcoOptiGen outperforms the best untuned GPT-3.5 model in the HELM benchmark. To verify whether the performance gain is simply due to the increased number of responses, we add a 'HELM (modified)' method in Table 2 which modifies the number of responses to match the inference budget consumed by the best configuration from EcoOptiGen. We also add 'EcoOptiGen (HELM budget)' which is EcoOptiGen's performance when using the same inference

Table 3: Tuned results using different GPT-3.5 models for $B.i = 2K, B.o = 1M$. * indicates the model is the best for that dataset according to HELM.

| Model | APPS | HumanEval | MATH | XSum |
|---|---|---|---|---|
| code-davinci-002 | 0.07* | 0.819* | 0.856 | **0.198** |
| text-davinci-002 | 0.20 | 0.847 | 0.785 | 0.144* |
| text-davinci-003 | **0.21** | **0.861** | **0.863**$^*$ | 0.119 |

budget as HELM per task. The comparison between the first two rows or between the last two rows of Table 2 suggests that the hyperparameters in the HELM benchmark are under-tuned, and jointly tuning all the hyperparameters can be better than simply increasing the number of responses.

In Table 2, we also compare EcoOptiGen's performance with the other methods using hyperparameter search. We can see that with pruning, EcoOptiGen consistently outperforms the other non-pruning methods, and by a large margin on APPS and HumanEval. This confirms the effectiveness of the pruning technique. We further compare the number of trials searched by EcoOptiGen and the other search methods in Figure 5 of the appendix. We can observe that EcoOptiGen searches for 2-27× more trials under the same optimization budget, which helps it achieve the better result across all the tasks.

This study finds that, (a) compared to directly using the best evaluated configuration from HELM, one can potentially find much better configurations for a particular application by tuning the inference hyperparameters; and (b) pruning can vastly boost the optimization efficiency.

### 4.3 Effect of Inference Budget

To understand how the performance of EcoOptiGen is affected by the inference budget, we further vary the inference budget $B.i$ from 500 tokens to 2000 tokens on APPS, HumanEval and MATH, while fixing the total optimization budget $B.o = 1M$. Table 5 in the appendix displays the performance scores and Figure 6 displays the number of trials finished within the optimization budget. On APPS, the average number of input tokens is larger than 500, so no performance score is available in that case. On HumanEval, the optimized performance score increases from 0.653 to 0.819 as the inference budget increases from 500 to 2000. On MATH, the performance score increases from 0.398 to 0.863 as the inference budget increases from 500 to 2000. On APPS, the performance score drops from 0.10 to 0.07 when the inference budget is increased from 1500 to 2000. Based on Figure 6, we hypothesize that the performance drop is due to the decrease of the number of trials within the total optimization budget as the average cost per trial increases. We perform an additional experiment to test that hypothesis: we increase $B.o$ to 2M for $B.i = 2000$ on APPS. The optimized performance score then increases from 0.07 to 0.12, and the number of trials increases from 86 to 165. The result supports the hypothesis.

The takeaway message in this study is that EcoOptiGen is able to find significantly better configurations with increased inference budget, unless the optimization budget is not enough.

### 4.4 Effect of Model

In previous experiments, we fixed the model on each dataset according to the HELM benchmark. In this subsection, we first apply EcoOptiGen to other models in the GPT-3.5 family on each dataset, using an inference budget 2K and a total optimization budget of 1M. At the time this experiment was conducted, it was generally recommended by OpenAI to use "code-davinci-002" for code generation and "text-davinci-003" for other text generation (Shieh, 2022).

Table 3 summarizes the results. On APPS, HumanEval and MATH, text-davinci-003 performs the best after tuning. On XSum, code-davinci-002 performs the best after tuning. On three datasets, the best models after tuning are different from the best models according to the HELM benchmark, as seen by the mismatches of the asterisk (HELM) and bold (after tuning) in each column. On

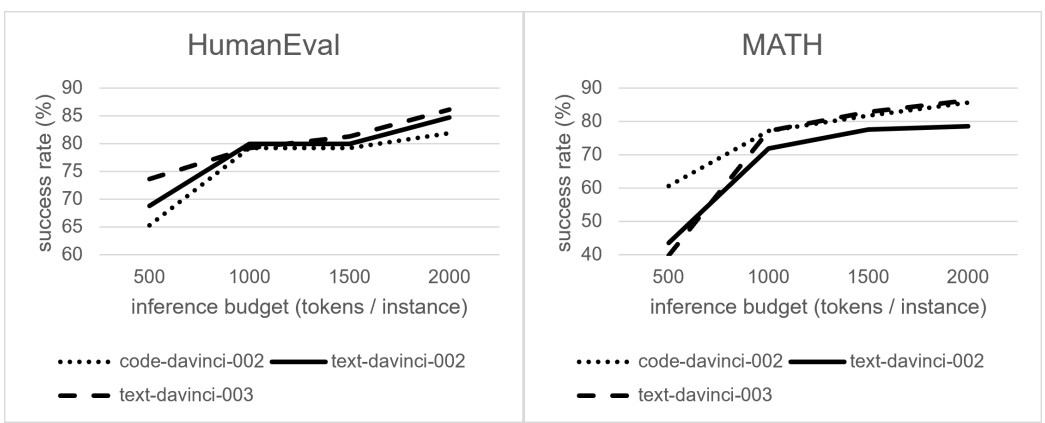

Figure 3: Results using GPT-3.5 models with varying inference budgets and a fixed optimization budget.

APPS and HumanEval, we find that text-davinci-003 excels in the two code generation tasks. On MATH, code-davinci-002 can get close to text-davinci-003 after tuning. On XSum, we find that code-davinci-002 surprisingly outperforms the text-davinci models by a large margin. These results are quite contradictory with the common beliefs on model selection (Shieh, 2022).

We further plot the model performances with respect to each individual inference budget in Figure 3. We find that for HumanEval, text-davinc-003 model performs the best consistently. On MATH, although Table 3 shows that text-davinci-003 model slightly outperforms code-davinci-002, Figure 3 shows the latter is actually superior in the low inference budget range.

The takeaway message of this study is that with EcoOptiGen, the best performing model is not always the commonly recommended model. This reveals one of the benefits of hyperparameter optimization in avoiding suboptimal choices due to idiosyncrasies. A newer model is not certain to outperform an older one.

## 4.5 ChatGPT Models

Two chat-optimized models powering ChatGPT (gpt-3.5-turbo and gpt-4) are released after the initial version of this paper. We evaluate EcoOptiGen's performance on them in this section.

We use the MATH dataset for evaluation. The setup is different from the previous sections to add diversity in the evaluated scenario. We use "success_vote" to compare the majority voting result for each problem with ground truth, instead of "success" which checks whether at lease on response is correct. We use the prompt template: "{problem} Solve the problem carefully. Simplify your answer as much as possible. Put the final answer in \boxed{}." We use all the levels from level 2 to level 5 in the Algebra category. We perform tuning per level, with both gpt-3.5-turbo and gpt-4 in the search space of "model". The common belief is that gpt-4 vastly outperforms gpt-3.5-turbo in math problems OpenAI (2023). We compare with gpt-4 using default inference hyperparameters.

Figure 4 shows the average accuracy and average inference cost of each configuration. On Level 2, surprisingly, the tuned gpt-3.5-turbo model is selected as a better model and it vastly outperforms untuned gpt-4 in accuracy (92% vs. 70%) with equal or 2.5 times higher inference budget. The same observation can be obtained on Level 3. The selected model changes on Level 4 and 5. The tuned gpt-4 achieves much higher accuracy (56% vs. 44% on Level 4, 35% vs. 20% on Level 5) and lower cost than the untuned gpt-4. These results additionally verify the robust effectiveness of EcoOptiGen and reinforce the takeaways in the previous sections. The opportunity for performance tuning still exists with the continual advancement of LLMs.

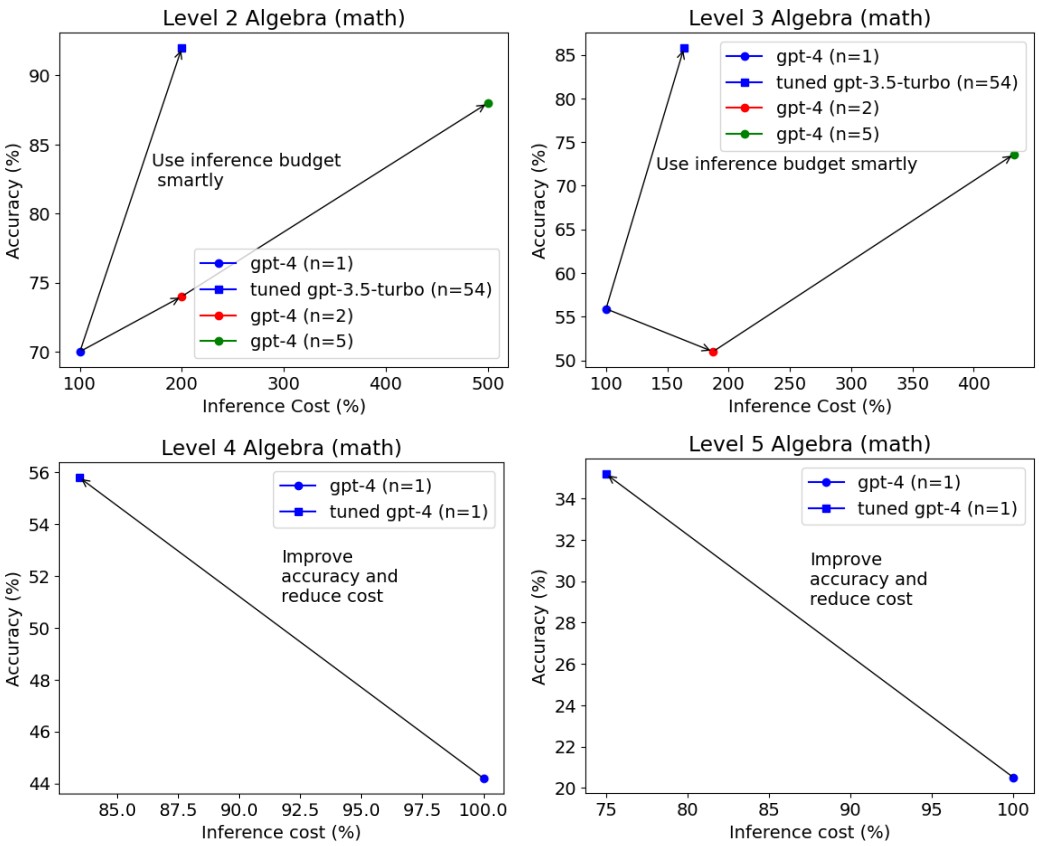

Figure 4: Tuning ChatGPT models for MATH.

## 4.6 Discussions and Future Work

For the summarization task the improvement by tuning is not as large as in code and math tasks. One potential reason is the ranking criterion for selecting one response from the *best_of* responses is not aligned with the final evaluation metric.

It will be interesting future work to develop methods that help with understanding the optimized hyperparameter choices (ref. Table 6 in the appendix). It is also possible to further automate the tuning. For example, the current solution takes user-specified choices of prompts as the input search space. Automatically searching for optimal numbers and choices of demonstration examples can potentially result in more effective ways of using the inference budget.

## 5 Related Work

Hyperparameter optimization methods for generic machine learning models have been studied for a decade (Feurer and Hutter, 2019; Bergstra et al., 2011). Since the training of deep neural networks is very expensive, new HPO methods have been proposed to reduce the cost required. Early stopping methods (Li et al., 2017) stop training with unpromising configurations at low fidelity (e.g., number of epochs) by comparing with other configurations' training performance at the same fidelity. Later, cost effective hyperparameter optimization were proposed. For example, in BlendSearch (Wang et al., 2021), an economical hybrid search strategy was proposed to handle heterogeneous evaluation cost and its effectiveness is demonstrated in fine-tuning a transformer model Turing-NLRv2. Automated hyperparameter optimization is also studied specifically for NLP tasks, e.g., fine-tuning BERT-like language understanding models (Liu and Wang, 2021) and neural machine translation systems (Zhang and Duh, 2020).

## 6 Broader Impact Statement

Users of LLMs should take environmental impact into consideration when they determine inference budget and optimization budget. Our work helps reduce the energy consumption by helping users find deployable configurations with a budget cap, which is otherwise difficult to find and may result in increased environmental impact. When the budget is not set carefully, the optimization process could cause excessive energy consumption.

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

Table 4: Configuration for the HELM baseline.

| Dataset | Config |
|---|---|
| APPS | {'model': 'code-davinci-002', 'prompt': '{input}', 'max_tokens': 600, 'temperature': 0.2, 'stop': ['''',– – –,'"""',\n\n\n]} |
| HumanEval | {'model': 'code-davinci-002', 'prompt': '{definition}', 'max_tokens': 600, 'temperature': 0.2, 'stop': [\nclass, \ndef, \nif, \nprint]} |
| MATH | {'model': 'text-davinci-003', 'prompt': 1-shot chain-of-thought demonstration*, 'max_tokens': 400, 'temperature': 0, 'stop': [###]} |
| XSum | {'model': 'text-davinci-002', 'prompt': 5-shot demonstration**, 'max_tokens': 400, 'temperature': 0.3, 'stop': [###]} |

* 1-shot chain-of-thought demonstration in MATH:

```
Given a mathematics problem, determine the answer. Simplify your answer as much as possible.
###
Problem: What is the value of $\sqrt{3! \cdot 3!}$ expressed as a positive integer?
Answer: $\sqrt{3!\cdot3!}$ is equal to $\sqrt{(3!)^2}=3!=3\cdot2\cdot1=\boxed{6}$.
###
Problem: {problem}
Answer:
```

** 5-shot demonstration in XSum:

```
###
Article: Almost one million people visited the city during the six-week festival period
over Christmas and Hogmanay. Organisers said almost 890,000 people visited the Edinburgh's
Christmas events in 2014/15, contributing \u00a3199.5m to the local economy. The three-day
Hogmanay celebrations attracted more than 150,000 people, creating an economic impact of
\u00a341.8m. Charlie Wood, Edinburgh's Christmas festival director, said: "This is great
news for Edinburgh. The revenue generated does not go to the events themselves, the event
organisers or to Edinburgh city council. "This is money, which is going to the businesses
of Edinburgh, be it retail, accommodation, food, drink, shopping and entertainment."

Summarize the above article in 1 sentence.
Edinburgh's winter festivals generated more than \u00a3241m for the city, according to
organisers.

###
Article: A firm in north Wales wants to bring the PooPrints service from the United States
to the UK with up to 15 councils reportedly interested in the scheme. Councils could make
owners in problem areas register their dogs to a database which involves a mouth swab taken.
Then, DNA could be taken from mess left on a street, path or grass and used to find a match
on the database. Gary Downie, managing director of Streetkleen Bio in Ruthin, Denbighshire,
believes local authorities can use new powers granted by the Antisocial Behaviour and
Policing Act 2014 to force dog owners to comply. "The purpose of the system is to get
cleaner, safer open spaces," he said. Councils the company is in talks with include
Kingston-upon-Thames in south-west London, Aberdeen and Cheshire East.

Summarize the above article in 1 sentence.
DNA in dog mess could be used to catch owners who fail to clear up their pet's mess.
```

**Algorithm 1** Configuration evaluator with pruning

---

1: **Inputs:** configuration $x$, tuning data $D$, target average number of tokens $B.i$, valid and invalid trials $X_{\text{valid}}$ and $X_{\text{invalid}}$.
2: **Outputs:** utility $U_x(D)$ if cost $C_x(D) \leq B.i$; 0 otherwise.
3: **Initialization:** obtain max_valid_n and min_invalid_n based on Eq. (1) and (2). $N \leftarrow x.n, n' \leftarrow 0, R \leftarrow []$
4: **if** $x.n \leq$ max_valid_n **then**
5:     $n \leftarrow x.n$
6: **else if** $x.n \geq$ min_invalid_n **then**
7:     return 0
8: **else**
9:     $n \leftarrow$ max_valid_n
10: **end if**
11: **while** True **do**
12:     $x.n \leftarrow n - n', k \leftarrow 1, k' \leftarrow 0$
13:     **while** True **do**
14:         **for** $i \in [k', k)$ **do**
15:             get responses from LLM for data point $d_i$ using configuration $x$ and append to $R$
16:         **end for**
17:         $\rho \leftarrow \begin{cases} (1 - \frac{k}{|D|})(1 + \frac{1}{k}) & 2k > |D| \\ 1 - \frac{k-1}{|D|} & 2k \leq |D| \end{cases}$
18:         **if** $C_x(D_k) > B.i(1 + .1\sqrt{\frac{\rho}{k}})$ **then**
19:             update $X_{\text{invalid}}$ and return 0
20:         **end if**
21:         **if** $C_x(D_k) \leq B.i(1 - .1\sqrt{\frac{\rho}{k}})$ **and** $(n < N$ **or** $k = |D|)$ **then**
22:             update $X_{\text{valid}}$
23:             **if** $n < N$ **then**
24:                 $R \leftarrow [], n' \leftarrow 0$, break
25:             **end if**
26:         **end if**
27:         **if** $k < |D|$ **then**
28:             $k \leftarrow \min(2k, |D|)$
29:         **else**
30:             break
31:         **end if**
32:     **end while**
33:     **if** $n < N$ **then**
34:         **if** $R \neq []$ **then**
35:             $n' \leftarrow n$
36:         **end if**
37:         $n \leftarrow \min(2n, N)$
38:     **else**
39:         Return $U_x(D)$
40:     **end if**
41: **end while**

---

###
Article: The works at Nottingham Castle include a chalk portrait of St Anne, sketches of bodies and plants, plus some technical drawings. The artist made only around 20 paintings during his lifetime, including the Mona Lisa and The Last Supper, but left many more drawings. In total, there are almost 600 drawings by da Vinci in the Royal Collection. They were originally bound into a single album, thought to have been acquired in the 17th Century by Charles II. Experts believe Leonardo's drawings are the richest, most wide-ranging and most technically brilliant of any artist. The exhibition is on show at Nottingham Castle Museum and Art Gallery until 9 October.

Summarize the above article in 1 sentence.
Rare drawings by Leonardo da Vinci, which are part of the Queen's royal collection, have gone on show.

###
Article: Distill Ventures, which is part of the Diageo group, said it was investing an unspecified sum in Melbourne-based Starward Whisky. This marks the second whisky investment for Distill, which was set up to back early-stage brands and help them grow. Last week, it announced investment in Denmark-based Stauning Whisky. David Gates, Diageo's global head of premium core spirits, said: "Australian whisky has rightly been gaining increasing global recognition recently and Starward has developed a uniquely positioned whisky to capture this opportunity." Frank Lampen, co-founder of Distill Ventures, added: "The Starward team are exactly the types of entrepreneur we love working with. "Their vision for the future is really exciting and this investment will enable increased production of their signature single malts and continued development of their innovation pipeline." Last year Diageo had a 37%% share of the Scotch whisky market in terms of volumes.

Summarize the above article in 1 sentence.
Diageo, the world's biggest Scotch whisky distiller, has invested in an Australian distillery to help it expand into new export markets.

###
Article: It follows reports of dog fouling and damage at the Camperdown and Caird Park courses. Dogs can still be walked across the courses but not if owners are playing a round of the game at the time. A spokesman for Leisure and Culture Dundee said the rules were changed on 20 April. He said: "This change reflects the concerns of many players and staff about dog fouling and damage being caused to the courses, particularly greens and bunkers. "The new management rules, which do not affect the Right to Roam legislation, are clearly signed at the courses and on the Leisure and Culture Dundee website. "Most golf courses in Scotland do not allow players to bring dogs with them."

Summarize the above article in 1 sentence.
Golfers at Dundee's public courses have been banned from bringing their dogs with them after complaints from fellow players and staff.

###
Article: {input}

Summarize the above article in 1 sentence.

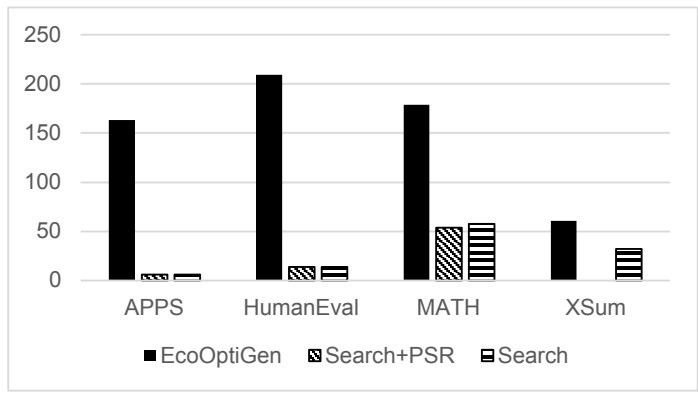

Figure 5: Number of trials within the total optimization budget.

Table 5: Results using varying inference budgets ($B.i$), with a fixed total optimization budget ($B.o$).

| $B.i$ ($B.o = 1M$) | APPS | HumanEval | MATH |
|---|---|---|---|
| 500 | - | 0.653 | 0.398 |
| 1000 | 0.05 | 0.792 | 0.771 |
| 1500 | 0.10 | 0.792 | 0.828 |
| 2000 | 0.07 | 0.819 | 0.863 |

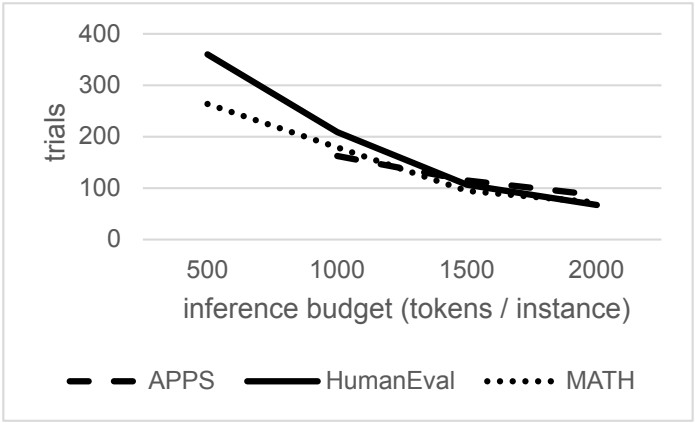

Figure 6: Number of trials with respect to varying inference budgets ($B.i$), with a fixed total optimization budget ($B.o$).

Table 6: Optimized hyperparameter configurations for text-davinci-003 with $B.i = 2K$, $B.o = 1M$.

| Task | max_tokens | temperature_or_top_p | n |
|---|---|---|---|
| APPS | 176 | top_p: 0.982 | 15 |
| HumanEval | 517 | top_p: 0.682 | 18 |
| MATH | 193 | temperature: 1 | 26 |

