# OpenReview forum: "Cost-Effective Hyperparameter Optimization for Large Language Model Generation Inference"
_automl.cc/AutoML/2023/Conference — AutoML 2023 MainTrack_

### Review · Reproducibility_Reviewer_coVN · 2023-03-29

**Completeness Of Code And Dataset Supplement Rating:** 3
**Usability And Ease Of Reproducibility Rating:** 3

**Actions Required To Increase The Reproducibility And Overall Recommendation:**

Try downloading the code from the linked repo on a fresh computer or docker container, and then try following the given instructions.

**Completeness Of Code And Dataset Supplement:**

Unable to reproduce the results and get the code running. I followed the link in the paper to the code supplement: https://anon-github.automl.cc/r/LLM/ecooptigen/README.md

I was unable to install the required packages. I downloaded the repo from the link, but the XXXX-2 package had been renamed in the download which is what I think is causing the problems. I followed the instructions in the readme to install via requirements.txt and got the following result:
pip3 install -r requirements.txt
Collecting crfm-helm==0.1.0
  Downloading crfm_helm-0.1.0-py3-none-any.whl (1.2 MB)
     ━━━━━━━━━━━━━━━━━━━━━━━━━━━━━━━━━━━━━━━━ 1.2/1.2 MB 12.4 MB/s eta 0:00:00
ERROR: Could not find a version that satisfies the requirement XXXX-2[blendsearch] (from versions: none)
ERROR: No matching distribution found for XXXX-2[blendsearch]

Replacing XXXX-2 with the renamed package did not resolve this issue. I think the readme would greatly benefit from instructions beyond "pip install -r requirements.txt".

**Overall Reproducibility Review:**

It is hard to assess the reproducibility of the paper since I was not able to get the code to run. I think there might be an issue with downloading the repo since the UI on the website hosting the code has a directory called XXXX-2 which is not located in the download, although I cannot be sure what the issue was.

**Review Confidence:**

2: You are willing to defend your assessment, but it is quite likely that you did not understand the central parts of the submission or that you are unfamiliar with the code or data.

**Review Rating:**

3: Reject, you were not able to reproduce some critical aspects of the paper, and question whether it would be possible even with additional effort.

**Review Summary:**

I was unable to run the code.

**Summary Of Necessary Code And Dataset Supplement:**

The authors present EcoOptiGen which leverages hyperparameter optimization and cost-based pruning to maximize the value of outputs from LLMs (such as GPT-3.5) under a limited inference budget. Experiments are run with GPT-3.5. The authors had to implement this method which optimizes the hyperparameters (number of responses, temperature, max tokens, etc.) collectively and evaluate the method on several datasets which represent a variety of applications that LLMs might be used for.

Experiments on the method were then conducted:
1. How much gain in performance can the method get under a budget constraint?
2. How does varying the budget affect the performance?
3. How does varying the model affect the performance?

**Usability And Ease Of Reproducibility:**

I was not able to run the code. I would suggest more in-depth instructions.

---

### Official Review · Reviewer_fssk · 2023-03-29

**Potential Impact On The Field Of Automl Rating:** 3
**Technical Quality And Correctness:** I am pretty confident in the presente…
**Technical Quality And Correctness Rating:** 4
**Clarity Rating:** 3
**Actions Required To Increase Overall Recommendation:** NA

**Summary Of Contributions:**

How to tune the hyperparameters or search for the architecture of Large Language Models is one of the very contemporary questions, considering the size of these models and the currently heuristically & manually driven search. This paper provides a first, very targeted step into a more feasible subproblem of the hyperparameter search surrounding LLMs. Particularly, they concern themselves with inference-related hyperparameters of GPT models and try to optimize the utility of the generated responses on a test set of queries (context) used for tuning their search space. They tend to this problem in a cost-constrained setting, where each response generated under a context and evaluated for a given hyperparameter setting has monetary implications. A software developer trying to optimize the usefulness of the generated text of his application very well would want to cut down on the hyperparameter tuning cost.
They clearly demonstrate that tuning hyperparameters jointly considerably can increase performance (as one would expect).

From an AutoML perspective, they use a local-search-augmented Bayesian Optimization setup called BlendSearch, which they adapt to their cost-constrained setup. They similarly do this to aggressive racing over instances, but adapted according to use case-specific properties of their search space and problem.

**Clarity:**

I found section 3.2. very cumbersome to read. Would highly recommend making it clearer from the get-go, that this evaluation procedure is trying to race the number of evaluations for a configuration and what assumption is made to do this, before delving into the details. This section reads as badly as the associated algorithm 1.

I think it would be helpful to amend paragraphs 67-86 with a few more details or examples on how these hyperparameters affect the performance.

**Overall Review:**

Weakness:
1. Noticeably, - and as the authors admit in the discussion section "Automatically searching for optimal numbers and choices of demonstration examples can potentially result in more effective ways of using the inference budget."
the distribution of "instances" i.e. user demonstrations needs to be well crafted for the end application which is a hindrance to their applicability.

**Potential Impact On The Field Of Automl:**

As stated above LLMs are a very hot topic. Although this is not a method to find or improve a new LLM variant, it surely is the first low-hanging fruit for developers who want to build downstream applications and are cost-constrained. As LLMs are likely not solved with a single solution, this surely is an important step to compartmentalize the LLM search space. This paper on its own is at least a good first step in this direction. Formulating their search space solely on GPT derivatives is likely somewhat limiting, but I am confident, that it is relatively easily adaptable to other LLM models.


**Review Confidence:**

4: You are confident in your assessment, but not absolutely certain. It is unlikely, but not impossible, that you did not understand some parts of the submission or that you are unfamiliar with some pieces of related work.

**Review Rating:**

7: Weak Accept: Technically sound paper with moderate-to-high impact and strong evaluation, with perhaps some minor flaws.

**Review Summary:**

As outlined above I appreciate the divide-and-conquer strategy on LLM models and the very targeted & feasible approach the authors take here. This paper will likely spawn discussions in this area. I would hope for a little more than applying well known BO method with local search in a cost-constrained racing heuristic (which is why I suggest a weak accept), but this is still a valid first step in dealing with LLMs from an AutoML perspective.

---

### Official Review · Reviewer_bzx5 · 2023-04-07

**Potential Impact On The Field Of Automl Rating:** 3
**Technical Quality And Correctness Rating:** 3
**Clarity Rating:** 3
**Actions Required To Increase Overall Recommendation:** Any action that addresses any of the …

**Summary Of Contributions:**

The paper presents a study on hyperparameter optimization for text generation inference using Large Language Models (LLMs). It proposes a cost-effective hyperparameter optimization method named EcoOptiGen, which optimizes multiple hyperparameters jointly, including the number of responses, max tokens, temperature, probability mass, and prompts. The paper also introduces a cost-based pruning strategy to improve the optimization efficiency under budget constraints. The effectiveness of EcoOptiGen is evaluated on four datasets, and it is shown that EcoOptiGen can find higher quality hyperparameter settings than the default settings suggested by a recent LLM benchmark. The paper shows that the holistic hyperparameter optimization can prevent suboptimal results and provides evidence that a tuned alternative model can outperform the recommended model.

**Clarity:**

In the introduction, it should be made clear that the objective of the study is to determine the usefulness of searching for inference hyperparameters in the context of Large Language Models (LLMs). Research questions should also be included in the introduction.

Table 2 is not presented as clearly as it could be. It is not clear whether the horizontal line that divides the approaches separates different inference budget. Additionally, has HELM (modified) the same inference budget as EcoOptiGen?

Is the optimization budget a limitation of the work?

It would be beneficial to have a conclusions section that summarizes the key findings of the study and provides suggestions for future research directions.

**Ethics Details (Optional):**

No ethical concerns. However, I give notice that the paper was uploaded to arxiv (https://arxiv.org/abs/2303.04673) so that the preprint format option can be checked if necessary.

**Overall Review:**


Strengths:

The paper addresses a novel and important issue of exploring inference hyperparameter space for Large Language Models (LLMs).
Proposed EcoOptiGen method, is novel and provides a hyperparameter optimization framework for LLMs.
The paper demonstrates the effectiveness of the EcoOptiGen method on multiple datasets and shows that it can find higher quality hyperparameter settings than the default settings suggested by HELM.
The paper presents a cost-based pruning strategy that improves optimization efficiency under budget constraints.
The findings of the paper are important for researchers and practitioners who work with LLM-enabled applications.

Weaknesses:

The paper could have provided a more detailed explanation of the BlendSearch hyperparameter search method and its appropriateness for the current problem.
The paper could have presented more clearly analysis to make it easier for readers to understand the differences between the approaches, inference budget implications, and possible errors that occurred during optimization, which could have helped to understand the problem better.
The paper does not perform statistical significance tests for the results.





**Potential Impact On The Field Of Automl:**

The paper's contributions are novel and valuable for the field of AutoML. It presents a practical optimization method that can find higher quality hyperparameter settings than the default settings suggested by recent HELM benchmark. Researchers who work on developing LLM-enabled applications and optimizing their performance can benefit from the proposed EcoOptiGen method. The paper's ideas and findings can also help guide future research in the field of AutoML.

**Review Confidence:**

4: You are confident in your assessment, but not absolutely certain. It is unlikely, but not impossible, that you did not understand some parts of the submission or that you are unfamiliar with some pieces of related work.

**Review Rating:**

7: Weak Accept: Technically sound paper with moderate-to-high impact and strong evaluation, with perhaps some minor flaws.

**Review Summary:**

Based on my analysis of the paper, I recommend that the paper be accepted. The paper addresses an important and timely issue of AutoML concerning Large Language Models (LLMs). The proposed EcoOptiGen method is novel and preliminary results demonstrates that it can find higher quality hyperparameter settings than the default settings suggested by recent LLM benchmarks.

**Technical Quality And Correctness:**

The choice of the BlendSearch hyperparameter search method is not entirely clear. Is the choice of BlendSearch enough to say that the search is cost-effective? Comparisons with other search methods would clarify this claim.

It appears that the greater the inference budget, the higher the success rate by the optimizer and the fewer number of attempts needed. It is not entirely clear why this happens. Shouldn't a drop be seen for all tasks when the budget is too big?

A categorization of the possible errors would be useful in understanding the problem. What types of failures can we expect from LLMs and how does it relate to the success rate achieved?

There is no analysis of the statistical significance of the results.

---

### Official Review · Reviewer_1igu · 2023-04-09

**Potential Impact On The Field Of Automl Rating:** 4
**Technical Quality And Correctness Rating:** 3
**Clarity Rating:** 3

**Summary Of Contributions:**

This paper presents a novel method for optimizing the inference hyperparameters of large language models (LLMs). The approach involves a cost-based pruning strategy paired with a black-box optimization technique, namely BlendSearch.

The authors describe a dynamic pruning approach that eliminates regions of the search space where further text generation would exceed the allocated cost/utility. The method also iteratively doubles the number of samples while ensuring the cost does not exceed a pre-defined budget, reducing the linear task into a logarithmic scale.

The proposed approach is evaluated using the HELM protocol and compared with similar strategies. The results demonstrate the superiority of the proposed approach, with further ablation studies supporting the claim.

**Actions Required To Increase Overall Recommendation:**

In my overall review, I suggested that running the proposed approach with different examples sampled each time as the tuning dataset would increase the confidence in the results and investigate the scalability of the approach. Additionally, the authors should justify the cost of performing the tuning strategy against the minimal improvement achieved, such as 0.004 on XSum, 0.02 on APPS, and less than 0.1 on HumanEval.

**Clarity:**

The paper's contributions are well-defined, but some crucial aspects of the understanding are relegated to the appendix, such as the pseudo-code. Moreover, the BlendSearch optimization approach is only briefly introduced.

**Ethics Details (Optional):**

No ethical concerns

**Overall Review:**

The paper offers several strengths:

- The authors explore the significance of optimizing inference hyperparameters for language models, an increasingly expanding field. They demonstrate that a straightforward tuning strategy can enhance performance over default hyperparameters.
- The proposed pruning approach is both easy to understand and efficient.
- The authors successfully employ progressive sampling in order to reduce the chance of incorrect pruning.
- The suggested method surpasses other basic hyperparameter optimization methods, as well as the default hyperparameters recommended.

There are some concerns with the paper, including:

- As noted in the Clarity section, the rationale for using only 20 examples for tuning is not explained. Presenting the average performance of the tuning strategy across multiple sampling instances could help justify this approach further.

- The authors explore a limited search space and ignore some hyperparameters.

- The paper does not investigate the temporal aspect of using the proposed hyperparameter tuning strategy. It is not clear whether the minor improvement in performance on 75% of the datasets justifies the additional computational cost compared to using the default hyperparameters, which do not require any tuning.

**Potential Impact On The Field Of Automl:**

The approach presented in this paper is the first to address the problem of tuning inference hyperparameters and demonstrates the effectiveness of an established tuning strategy paired with a novel pruning approach on the overall quality of text generation.

**Reproducibility (Optional):**

The paper provides an associated code repository.

**Review Confidence:**

4: You are confident in your assessment, but not absolutely certain. It is unlikely, but not impossible, that you did not understand some parts of the submission or that you are unfamiliar with some pieces of related work.

**Review Rating:**

8: Accept: Technically sound paper with major impact and strong evaluation, with perhaps some minor flaws.

**Review Summary:**

The paper addresses the significance of tuning inference hyperparameters for large language models and proposes a practical and effective tuning strategy. By utilizing progressive sampling of dataset examples and BlendSearch implementation, the authors demonstrate the quality of their approach over default hyperparameters on four datasets. This study fills a crucial gap in the rapidly growing field of generative AI.

**Technical Quality And Correctness:**

The authors evaluate the proposed method on multiple datasets using the HELM protocol, but the choice of using only 20 tuning samples per dataset is unclear as the rationale for this is not explained. Moreover, the authors do not discuss the ability of their approach to generalize when using different tuning instances.

---

### Official Review · Reviewer_FqAX · 2023-04-12

**Potential Impact On The Field Of Automl Rating:** 4
**Technical Quality And Correctness Rating:** 3
**Clarity Rating:** 2

**Summary Of Contributions:**

&nbsp;

The authors introduce EcoOptiGen, a black-box optimizer based on BlendSearch for inference hyperparameter optimization of LLMs. The scheme features a pruning strategy to filter invalid hyperparameter configurations that exceed a cost budget based on token count numbers for the LLM. The main experimental findings are:

&nbsp;

1. EcoOptiGen outperforms untuned LLM baselines on new tasks, indicating that there are performance gains to be achieved through optimization of LLM inference hyperparameters.

2. The authors perform a a series of experiments with definitive conclusions e.g. increasing the token budget improves the performance of EcoOptiGen.

&nbsp;

**Actions Required To Increase Overall Recommendation:**

&nbsp;

1. Address reproducibility concerns stemming from the checklist answers (1 point raise).

2. Conditional on addressing the reproducibility concerns I will upgrade my score by a further point (2 points in total).

&nbsp;

**Clarity:**

&nbsp;

Below I list the issues I experienced with clarity. I believe all of these points are addressable and as such, will increase my score as and when they are addressed.

&nbsp;

**__MAJOR POINTS__**

&nbsp;

1. It would be very helpful to the reader if a concrete problem definition was included. It would be great if this was provided together with the notation at the beginning of section 2. As it stands, I had to skip to later sections in order to understand the problem definition. An example of one clarification that could be made in the background section is the precise form of the inference and optimization budget constraints; i.e. for the inference budget constraint, an equation specifying some convex combination of the amount of prompts the user can supply to the LLM and the max tokens in dollar cost would be helpful in understanding the setting. A second figure could be included alongside Figure 1 at the top of page 2 in order to aid in explaining the setup.

&nbsp;

**__MINOR POINTS__**

&nbsp;

1. Typo for the citation on line 69/92 (ope)?

2. The appendix has no headings and as such, it is difficult to read the information.

3. It might be helpful if the code-style variable notation i.e. D.in, B.i and B.o was translated to mathematical notation in order to abstract the problem framework.

4. It would be great if Figure 4 had a vertical axis label.

6. The piecewise function defined on line 17 of Algorithm 1 in the appendix should include "if" in both lines to indicate the conditions.

7. Line 12 of Algorithm 1 in the appendix, would it make sense to switch the order of k and k' for readability?

8. Lines 18/21 of Algorithm 1 in the appendix, 0.1 would be clearer than .1.

9. Line 72, "words or word pieces" do the authors mean characters or subwords?

10. For clarity it would help if the "Tuning Data" could be further subdivided into the data that is used to tune the inference hyperparameters and the data that is used for independent evaluation i.e. the test data that is referred to in the experiment setup section. From the description on page 3, the reader does not know which is which.

11. Line 100, "For example, the 100 HumanEval (Chen et al., 2021) dataset contains an input field which is the concatenation of the 101 Python function signature and the doc string, and a test field of the test code." It would be great if at this stage in the paper, the HumanEval task could be provided i.e. the task is to generate code given the signature and docstring as input.

12. Table 1, type in the Note column, "will be replaced".

13. For the notation in Figure 2, is the utility a function of x or D? As far as I understand the tuning data D remains fixed and thus should be a fixed parameter of the utility U and the cost C. On the other hand, x, the hyperparameter configuration is the dependent variable and thus it may be more notationally consistent to use U_D(x) and C_D(x)? In the case of progressive subsampling, U and C will indeed be functions of D as well.

14. It would be great if some background on BlendSearch could be provided as this would make details of the algorithm clearer e.g. what was the surrogate model for the Bayesian optimization component of BlendSearch?

15. A corollary of the Hoeffding-Serfling inequality appears to be used in Algorithm 1, line 17. It would be great if this component of the algorithm was explained in further detail.

16. "For each dataset, we randomly sample a few hundred examples for testing". It would be great if the exact numbers could be provided.

17. In the caption of Table 2 it would be good to include what the metrics reported are.

&nbsp;


**Overall Review:**

&nbsp;

**__MAJOR POINTS__**

&nbsp;

1. In terms of suggesting a formal problem statement. It should be possible to frame the current problem as one of black-box optimization under unknown constraints [1], $\argmax  f(x)  s.t.  C(x) < Budget$ where f is an objective function measuring the utility of the LLM output and x is a vector representing the LLM inference hyperparameters (configuration). In the case of code generation for example this would be the supplied function signature and docstring. In this setting, the cost constraint is unknown but can be learned using a separate surrogate model to the surrogate for the utility. While learning the unknown constraint is simply an idea for future work, it would be good to consider how the LLM hyperparameter inference procedure fits into the unknown constraint framework.

2. For the baselines, could random search over the hyperparameter configurations be added as a baseline?

&nbsp;

**__MINOR POINTS__**

&nbsp;

1. In the introduction, it may be worth citing [2, 3] for some of the limitations of ChatGPT on math problem solving. With the current merge with Wolfram Alpha forthcoming it may be important to highlight that LLMs as a piece of standalone machinery still have great difficulties solving graduate level mathematics problems.

2. It appears as though much of the code in the repository comes from [4]. This is understandable as the method is used and acknowledged as a component of the paper. It would, however, be appropriate to mention explicitly in the README of the authors' codebase that portions of the code are derived from the codebase of Wang et al.

&nbsp;

**__REFERENCES__**

&nbsp;

[1] Gelbart et al. [Bayesian optimization with unknown constraints](http://auai.org/~w-auai/uai2014/proceedings/individuals/107.pdf). In 30th Conference on Uncertainty in Artificial Intelligence, UAI 2014 (pp. 250-259). AUAI Press.

[2] Frieder et al. 2023. [Mathematical capabilities of ChatGPT](https://arxiv.org/abs/2301.13867). arXiv preprint arXiv:2301.13867.

[3] Borji, A., 2023. [A categorical archive of ChatGPT failures](https://arxiv.org/abs/2302.03494). arXiv preprint arXiv:2302.03494.

[4] Wang et al. 2021. [Economic hyperparameter optimization with blended search strategy](https://openreview.net/forum?id=VbLH04pRA3). In International Conference on Learning Representations.

&nbsp;


**Potential Impact On The Field Of Automl:**

&nbsp;


Optimization of LLM inference hyperparameters is a highly topical and societally important problem given the near prospect of LLM integration into research and industrial workflows. At the time of writing, the number of prompts in GPT-4 is limited to 25 every 3 hours due to resource constraints. As the authors state, environmental considerations will ensure that LLM inference efficiency will remain to be a concern into the future. Given that this paper is one of the first to consider optimization of LLM inference hyperparameters, I believe that it will not only be impactful for the AutoML community yet will also serve to champion the benefits of AutoML for LLM practitioners.

&nbsp;


**Reproducibility (Optional):**

&nbsp;

1. In reference to the submission checklist there is a note that the raw results themselves and the code used to generate figures and tables based on the raw results is not provided. This is a major concern for reproducibility. It would be great if the authors could address this point in the rebuttal.

2. I attempted to run the code and it appears as though the requirements.txt file doesn't have the necessary dependencies?

&nbsp;

**Review Confidence:**

5: You are absolutely certain about your assessment. You are very familiar with the related work and checked all the details carefully.

**Review Rating:**

5: Borderline Leaning Reject: Technically sound paper where reasons to reject nonetheless outweigh reasons to accept. Please use sparingly.

**Review Summary:**

&nbsp;

The authors tackle a topical problem in this paper. If the issues on clarity and reproducibility can be addressed I will strongly consider raising my score.

&nbsp;

**Technical Quality And Correctness:**

&nbsp;

Issues are subsumed by the reproducibility issues described below. One constraint the authors were subject to when writing this paper is lack of control over the LLM APIs which impacted e.g. the implementation of true random trials. It would be great to see discussion of this issue during the review process.

For the time being I give a score of 3 on this point because it is still unclear how much control the authors had in relation to the empirical setup.

&nbsp;

---

### Official Review · Reviewer_7kJp · 2023-04-14

**Potential Impact On The Field Of Automl Rating:** 3
**Technical Quality And Correctness Rating:** 4
**Clarity:** This paper is well written and very e…
**Clarity Rating:** 3

**Summary Of Contributions:**

This paper focuses on optimizing inference hyperparameters in Large Language Models (LLMs) to maximize the value of text generation while operating under a limited inference budget. It introduces EcoOptiGen, a framework designed for economical hyperparameter optimization and cost-based performance evaluation. It uses existing method BlendSearch which combines Bayesian Optimization (BO) and local search for cost efficiency. Experiments show that by tuning the inference parameters such as the number of responses, temperature, and max tokens on the utility and cost of text generation in LLMs, the GPT-3.5 can achieves better performance on several benchmarks compared to the default parameters.

**Actions Required To Increase Overall Recommendation:**

In general, I feel this paper conducts a good empirical study on how to leverage existing blackbox optimiation and small set of data for hyperparameter tuning. Although I believe this is an important problem for industry or LLM users, the current presentation is lack of technical novelty and cannot demonstrate the challenge in this task. Instead it leverages existing algorithms to build the system easily. Moreover, this paper does not fully justify the design choice - lack of comparisons with some other blackbox optimiation approaches. Finally, there are no insights or explanation provided for the optimized parameters.

**Overall Review:**

Strengths:
- This paper is well written and easy to follow. The formulation is clear: optimizating the performance on the validation set within specific budget.
- The paper is well motivated and tackles an important problem - inference parameter optimiation for LLMs.

Weaknesses:
- Limited technical novelty. The challenges of the problem is not clear.
- The design choices are not sufficiently demonstrated with experiments. More optimization baselines should be considered. The proposed approach is not demonstrated on other non GPT-3.5 models.

**Potential Impact On The Field Of Automl:**

This paper raises an important problem of how to enhance the LLM's performance by optimiation the inference parameters. LLM has attracted a lot of attention recently and leads to tremedous success. In practice this project is useful for the community and industry.

**Reproducibility (Optional):**

This paper provides the code for reproducibility. The experiments in the paper are not difficult to reproduce.

**Review Confidence:**

3: You are fairly confident in your assessment. It is possible that you did not understand some parts of the submission or that you are unfamiliar with some pieces of related work.

**Review Rating:**

4: Weak Reject: For instance, a paper with minor technical flaws, limited impact, and/or weak evaluation.

**Review Summary:**

This paper has limited technical novelty and the emipirical study is not comprehensive (although I believe this is a nice project that tackles important problems).

**Technical Quality And Correctness:**

This proposed approach is easy to understand and the experiments are convincing. However, there are no experiments to verify the design choices. Moreover, this paper only consider GPT-3.5 models and it is not clear if the proposed method is working for all LLMs.

---

### Comment · Program_Chairs · 2023-05-16
**Additional reproducibility review**

There seem to be some problems with Anonymous GitHub which could have been resolved by the authors if they reacted to the repo review. In fact, the authors did anonymize some words using Anon GitHub, which broke their code and installation entirely! Moreover, no good instructions related only to the paper are given anywhere as far as I can tell. Besides, reproducing HPO for models used with these APIs is hard anyways as the reviewer will likely have no access to the API. This would perhaps been better to submit to the ABCD track in this state, the paper's work cannot be reproduced or verified (as the repo reviewer stated) but I think it is highly likely, considering the code and documentation quality, that it would have been usable without the messed up anonymization due to double blind. I also think the submission checklist could be improved by adding justifications for some of the points or correcting likely false statements (as reviewer FqAX also spotted).

In summary, in my opinion, there are issues with the reproducibility of the paper's contribution that are unnecessary and could have been resolved if the authors replied or put more effort into instructions and, for example, a minimal example. Without the anonymization this likely would have been easy to use as a python package. Still, as the reviewers with a higher rating mainly pointed out, this is a very interesting contribution and could add some valuable discussion/input to the community. Moreover, the final version public non-anonymize code will very likely not have reproducibility problems. Yet, we as a conference cannot attest the reproducibility of the actually used methods and results reported in the paper.